# Third Case of Visceral Leishmaniasis in COVID-19: Mini Review Article

**DOI:** 10.3390/pathogens11080913

**Published:** 2022-08-14

**Authors:** Claudia Colomba, Cristoforo Guccione, Raffaella Rubino, Michela Scalisi, Anna Condemi, Sara Bagarello, Salvatore Giordano, Antonio Cascio

**Affiliations:** 1Department of Health Promotion, Maternal and Infant Care, Internal Medicine and Medical Specialties, University of Palermo, 90127 Palermo, Italy; 2Division of Paediatric Infectious Diseases, “G. Di Cristina” Hospital, ARNAS Civico, 90134 Palermo, Italy; 3Infectious and Tropical Diseases Unit, AOU Policlinico “P-Giaccone”, 90127 Palermo, Italy

**Keywords:** visceral leishmaniasis, COVID-19, coinfection

## Abstract

Background: In the currently ongoing coronavirus pandemic, coinfections with unrelated life-threatening febrile conditions may pose a particular challenge to clinicians. Leishmaniasis is a zoonosis that may present general symptoms, including fever, malaise, and arthralgia, rendering it indistinguishable from COVID-19. Methods: In this paper, we aim to draw attention to this issue and analyze the clinical characteristics of the coinfection SARS-CoV-2/Leishmania through a systematic review of the literature. We were motivated by the observation of the first case of visceral leishmaniasis and COVID-19 in a paediatric patient. Conclusion: Our case is a reminder for healthcare providers to consider the diagnosis of visceral leishmaniasis in patients presenting with febrile syndrome in endemic regions during the COVID-19 pandemic.

## 1. Introduction

The ongoing COVID-19 pandemic represents an unprecedented global health challenge. Currently, health systems worldwide are engaged in unparalleled efforts, and physicians continue to play a critical role in the early detection and clinical management of the COVID-19 pandemic. COVID-19 infection in children is usually mild with low numbers of severe cases compared with adults [1].

However, in children, SARS-CoV-2 infection can frequently involve multiple body tracts, including the respiratory, gastrointestinal, musculoskeletal, and neurological systems [2,3,4]. In children, co-infections represent a particular challenge for clinicians because the current focus on the management of COVID-19 may, in part, lead to the inappropriate diagnosis and management of other life-threatening febrile medical conditions [5]. An appropriate epidemiological approach and a differential diagnosis are critical for selecting the most appropriate clinical intervention. Visceral leishmaniasis (VL) is an anthropozoonosis endemic in areas surrounding the Mediterranean Sea. In these areas, VL is caused by the protozoan *Leishmania infantum* and is transmitted through the bite of hematophagous sandflies belonging to *Phlebotomus* spp. Sicily is the largest island in the Mediterranean Sea, where VL is endemic [6]. In this paper, we describe the first report of a paediatric patient with VL who was also infected with SARS-CoV-2, and we analyze the clinical characteristics of the coinfection SARS-CoV-2/Leishmania through a systematic review of the literature.

## 2. Methods of the Review

A computerized literature review was performed using the PubMed and Scopus search engines by submitting the query: (leishman*) AND (COVID) OR (SARS-CoV) OR (nCoV-19) OR (coronavirus) OR (MIS-C) OR (PIMS-TS) to find cases of patients with VL and COVID-19 coinfection. No filters or language restrictions were applied to the results. Furthermore, all listed references were hand-searched for other relevant articles, and a citation tracker was used to identify any other relevant literature. An article was considered eligible for inclusion if it reported cases with complete clinical data consistent with Leishmania-SARS-CoV-2 coinfection.

## 3. Findings from Bibliography Research

From the literature search performed, we recovered 64 articles, 53 of which were excluded on the basis of their title or abstract. The remaining 11 were reviewed in full text. Among these 11, 10 were excluded because they did not report patients with concomitant VL and COVID-19. From the hand-search of the bibliography, we found one additional article. The bibliographic research process is schematised in Figure 1. In conclusion, two articles were selected, written by Miotti et al. and Pikoulas et al. [7,8].

The first case was reported in Italy and dated March 2020, during the first wave of SARS-CoV-2 [7]. This patient was an elderly man with several co-morbidities, including myasthenia gravis and selective erythroid aplasia, who was admitted for a sub-chronic fever and mild pancytopenia. VL was discovered. The patient became infected with SARS-CoV-2 during hospitalization and died just a few days later.

The second case was described in Greece in May 2021, probably during the Delta variant outbreak [8]. The patient was a young woman who had been admitted to a tertiary hospital in Athens for acute diarrhoea for 5 days and a fever the day before. On admission, she was tested for SARS-CoV-2, and the result was positive. In addition, a complete blood count showed pancytopenia and increased liver enzymes. Due to suspicion of neoplasia or infection, a bone marrow biopsy was performed and examined alongside a whole blood sample by molecular methods, which revealed positivity to *L. infantum*. After 13 days, during which treatment against Leishmania was administered, the patient was discharged with a remission of symptoms and clinical improvement.

## 4. New Case Report

A 4-year-old girl was admitted on 7 February 2022, to the Paediatric Emergency Department of the Children’s Hospital of Palermo, Sicily, for hyporexia and fever (peak 38°) that started 4 weeks earlier and an acute onset of chest pain. The child was the daughter of African migrants. However, she was born in Sicily, and the doctor denied that the child had ever been to her parents’ country of origin. During the physical examination on admission, the child appeared to be in relatively good clinical condition, slightly in pain, and the pharynx was hyperaemic without plaque or purulent secretion. Mild latero-cervical lymphadenopathy, mild hepatomegaly (2 cm below the costal margin), and severe hard splenomegaly (5 cm below the costal margin) were observed. No signs of respiratory or cardiac involvement were evident, nor were there any rashes, scabs, or oedemas.

Chest radiograph and electrocardiogram showed no alterations. Because of the evidence of hepatosplenomegaly, an ultrasound of the abdomen was performed showing a severely enlarged spleen (14.4 × 6.1 cm) and a mild ascitic flap.

A swab test for SARS-CoV-2 was performed in the hospital and was positive. Blood tests and a complete blood count revealed tri-linear pancytopenia with severe neutropenia (0.8 × 10^3^/mm^3^), severe thrombocytopenia (67 × 10^3^/mm^3^), and severe anaemia (Hb 6.6 g/dL, RBC 3.49 × 10^3^/mm^3^, Hct 19.7%, RDW-CV 21.3%, RDW-SD 43.8 fL). Other tests showed hyponatremia (132 mmol/dL), hypocalcemia (8.7 mmol/dL), hypoalbuminemia (3.3 g/dL), and elevations in LDH (256 IU/L), C-reactive protein (6.10 mg/dL), and alkaline phosphatase (117 IU/L). Other tests were normal.

Due to severe pancytopenia and SARS-CoV-2 infection, the child was admitted to the Infectious Diseases Department. On admission, a transfusion of concentrated red blood cells was administered, and because of splenomegaly and cytopenia, tests for Leishmania-DNA, a rapid test for malaria, and a serological assay for Leishmania were performed. All of these tests were positive for Leishmania and negative for malaria. Hence, amphotericin B therapy was started with the schedule of 3 mg/kg/day for 5 days, followed by the other dose on the 10th day. During hospitalization, the patient’s clinical conditions gradually improved and the Leishmania-PCR test turned negative. She was discharged to outpatient care to check on her whole blood count and provide a final administration of amphotericin B on the 10th day from the start of the therapy. 

Clinical progress, laboratory tests and the clinical management of the cases reported in this article are shown side-by-side in Table 1.

## 5. Discussion

Most children with SARS-CoV-2 infection develop no symptoms or mild symptoms, requiring only supportive care [9]. Coinfections with other pathogens, viruses, bacteria, and protozoa, should always be investigated, especially in cases, such as the one reported here, that present with a long-lasting fever [10,11,12]. In the diagnostic process of fevers of unknown origin, Leishmania infection should always be investigated in Sicily, the largest island in the Mediterranean and an endemic area for this zoonosis. It has been estimated that approximately half of the Sicilian population lives in areas at risk of Leishmania infection (rural areas, small villages or suburbs of cities), where sandfly vector species are more prevalent [6].

In contrast with the past, when VL was typically observed more commonly in children, the current age-related epidemiologic features observed in Sicily are consistent with reports from other Mediterranean regions of Europe, such as France, Spain, and Greece, where the ratio of childhood to adult cases is approximately 1:1 [13].

Typical clinical features of VL in children include fever, pallor, weakness, hepatosplenomegaly, and pancytopenia. As in our case, fever may be intermittent at first and then become continuous. Non-tender splenomegaly and hepatomegaly are caused by infection of the reticuloendothelial system. Pancytopenia caused by parasites invading the bone marrow is responsible for pallor due to anaemia and may subsequently cause haemorrhages due to thrombocytopenia and concomitant infections due to leukopenia. Anorexia and weight loss can lead to a wasting syndrome in misdiagnosed cases. Lymphadenopathy is found in some geographic areas, such as Sudan, but is less common in the Mediterranean region where it is an occasional finding unrelated to the disease [6,14]. Slightly latero-cervical lymphadenopathy was present in our case.

COVID-19, by contrast, is an emerging infection with clinical manifestations in children ranging from the most frequent mild cold to the rarest life-threatening, rapidly progressing, systemic disease that can lead to multi-organ failure [15]. During the current pandemic, the pathways of these two infections have inevitably crossed, with consequences that are not yet fully understood. Moreover, the early clinical symptoms (fever and asthenia) and laboratory parameters (e.g., leukopenia, thrombocytopenia, and elevated transaminases) might be similar in patients infected with SARS-CoV-2 and those with VL. Overlapping SARS-CoV-2 and Leishmania infection could lead to misdiagnosis, because leishmaniasis shares clinical and laboratory features with SARS-CoV-2 during the onset of the infection, making it difficult to determine which etiologic agent causes the disease.

One of the other two cases of Leishmania-COVID-19 coinfection reported in the literature is that described by Miotti et al. [7] in an immunocompromised adult patient. Protozoa persistence appears to be very common in infected individuals, and the development of an immunocompromised or immunosuppressive state affects various activities of innate and adaptive immunity in patients with VL. This is evidenced by the inability of peripheral blood mononuclear cells stimulated with leishmanial antigens to produce antibodies [16]. As a result, chronic infection may reactivate years after the initial contact with the parasite. The clinical presentation of VL in immunocompromised patients is similar to that observed in immunocompetent individuals. The main difference is the lower rate of response to treatment and the subsequent high rate of disease relapse [13,17,18,19,20].

In the case described by Miotti et al., although amphotericin therapy was well-tolerated and resulted in defervescence with a mild to moderate improvement in clinical conditions and laboratory tests, the concomitant VL and COVID-19, burdened by several comorbidities, was fatal.

The leishmaniasis-COVID-19 coinfection reported by Antonis Pikoulas may also have led to the reactivation of previously asymptomatic leishmaniasis [8]. Repolarization toward Th1 to deal with the virus may have allowed the parasite to escape immune surveillance, leading to symptomatic VL. There are many cases reporting that COVID-19 led to the reactivation of chronic, asymptomatic infections caused by viruses such as VZV (varicella-zoster virus), EBV (Epstein–Barr virus), CMV (cytomegalovirus), HSV (herpes simplex virus), HHV6 (human herpes virus 6), HBV (hepatitis B virus), protozoa, and fungi [21,22,23,24]. By contrast, VL may have led to a specific polarization of the immune response, making the patient more susceptible to viral infections, such as the widely circulating COVID-19, which was transmitted to the patient during the pandemic. Currently available data do not allow us to decide which scenario might be correct. Whatever the case, there are clear indications that the two infections likely result in complex immunologic interactions when their pathways cross [25].

Interactions between parasitosis and coronaviruses were extensively studied during the early stages of the pandemic, and the presence of a cross-talk between the two types of infection is confirmed. Researchers attempted to determine the relationships between the higher mortality observed in northern regions compared with Southern Europe and Africa with the lesser incidence of parasitosis and vector-borne diseases in the colder region. Indeed, following the first wave of COVID-19 in Europe in summer 2020, the possibility of herd immunity mediated by sandflies in Southern Europe was theorized by observing the distribution of arthropods and the lower impact of COVID-19 in these regions (most evident in Sicily, Cyprus, and Malta) [1,5]. Furthermore, an Iranian study carried out on 1.010 patients with a positive history of Cutaneous Leishmaniasis (CL) and 2.020 with no history of CL found a strong negative association between CL and severe COVID-19 [25]. Thus, despite the presence of an interaction between the immunological responses in these two conditions, the relevance of this needs to be better studied in the future.

Regarding laboratory diagnosis, we performed PCR on peripheral blood, a rapid and non-invasive method to diagnose immunocompetent children that is as sensitive as a diagnosis based on aspirated bone marrow. In addition, qualitative and semiquantitative PCR may be the standard method for monitoring treatment responses in immunocompetent children [26,27].

Liposomal amphotericin B (AmBisome) is nowadays considered the first-line treatment for VL. The efficacy and safety of a six-dose regimen of L-AmB is demonstrated by Cascio et al., who validated this first-line treatment for Mediterranean VL in children [28,29].

The current knowledge on the presentation of VL coinfection with COVID-19 could only be based on three reports. However, these three reports are for three different moments of the COVID-19 pandemic in Europe, with very different epidemiological characteristics (Figure 2). The first case was reported during the first wave, with a very high number of deaths per case, a poor control of COVID-19 spread, and no immunity in the population. The second case in Greece was reported during the Delta variant wave. The case reported here occurred during the Omicron outbreak, characterized by the highest ratio of cases to deaths because of the mild disease related to the dominant variant and the broad immunity of the population (by natural infection or vaccination). Hence, the evolution of the pandemic made COVID-19 cases less severe and testing positive for COVID-19 less important to define the origin of symptoms; therefore, positivity for COVID-19 could not lead to the aetiological diagnosis of clinical pictures and should not rule out other diagnoses.

To summarize, the present case is the third case of Leishmania-SARS-Cov2 coinfection, the first in a child, and the first during the Omicron variant. We highlighted the need for a thorough understanding of the relationship between neglected endemic diseases such as VL and pandemic SARS-CoV-2 infection. In particular, the complex immunology behind COVID-19 and rapidly changing pandemic scenarios force clinicians to deal with clinical conditions that are difficult to define at the first approach. Therefore, the complex interaction between parasites and the immunology of COVID-19 needs to be better explained in the future and forms an extremely interesting body of research that supports the “old friendly hypothesis” postulated by Rook [30]. This article also urges clinicians not to erroneously ascribe symptoms to COVID-19. Indeed, as for our patient, testing positive for SARS-CoV-2 could be incidental and could shift the focus away from an underlying life-threatening disease. Especially in the face of the Omicron epidemic, with a vast number of asymptomatic or paucisymptomatic cases and a low percentage of severe cases, clinicians should not immediately attribute signs and symptoms to COVID-19. However, differential diagnoses should be expanded, and any underlying, potentially life-threatening diseases should be identified.

## Figures and Tables

**Figure 1 pathogens-11-00913-f001:**
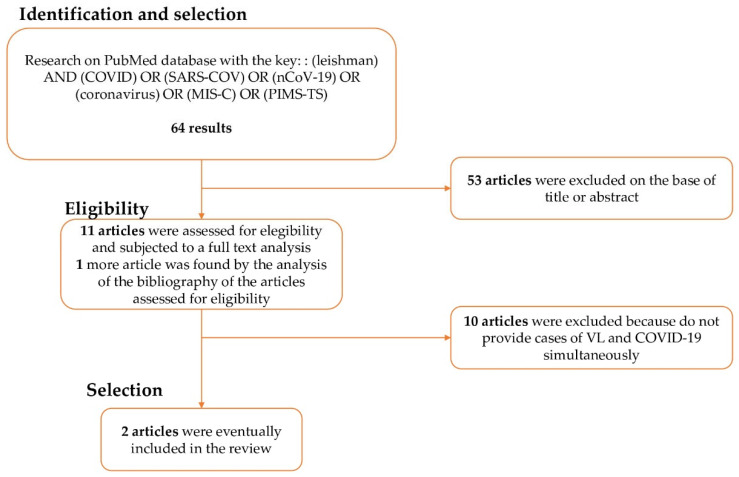
Process of bibliographic research.

**Figure 2 pathogens-11-00913-f002:**
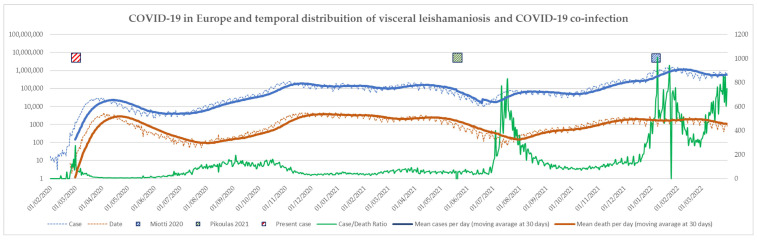
COVID-19 in Europe reports of coinfections. The figure shows: daily cases of COVID-19 infections, in blue; daily death due to COVID-19, in orange; the ratio between cases and death, in green; and the three case reports of VL and COVID-19. Curves describe cases and death.

**Table 1 pathogens-11-00913-t001:** Clinical progress and laboratory tests.

Author/Year	Age	Sex	Immunocompetence	Fever and Pancytopenia	Sample for Diagnosis	Diagnosis	Therapy	Outcome
**Miotti 2020**	70	M	No	Yes	Blood and bone marrow	Molecular diagnosis	Amphotericin B + dexamethason	Death
**Pikoulas 2021**	30	F	Yes	Yes	Blood and bone marrow	Molecular diagnosis	Amphotericin B + dexamethason	Full recovery
**Present case** **2022**	4	F	Yes	Yes	Blood	Molecular diagnosis	Amphotericin B	Full recovery

## Data Availability

All data used and/or analyzed during this study are included in this published article.

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
