# Peer review of "Third Case of Visceral Leishmaniasis in COVID-19: Mini Review Article"

_pathogens, 2022, doi:10.3390/pathogens11080913_

Round 1

Reviewer 1 Report

The authors performed a systematic review based on only 2 published articles and presented a clinical case. Although there is relevance to the topic, I discourage the manuscript to be sent as a systematic review. A case report with an good discussion, including the two cited articles, would be more appropriate.

Author Response

To the reviewer 1:

1) we actually conducted a literature review and discovered only two cases. As a result, we thought it was appropriate to change the title of the manuscript to: " Third case of visceral leishmaniasis and COVID-19. Mini review article.

2) we made only moderate English revisions as suggested

Reviewer 2 Report

Claudia Colomba and colleagues performed a revision on human visceral leishmaniosis and COVID-19 co-infection. The authors did a literature review, only identifying two cases and present themselves a new case of VL and COVID-19 co-infection. The manuscript is of scientific interest as it raises awareness to the COVID and other parasitosis co-infections problematic, addressing the challenges of diagnose and also laying foundation for future works in immunology of co-infections.

The manuscript is well written and easy to follow, and I only recommend to the authors to address this minor issues, before final acceptance.

Minor:

Line 13 – remove “with”

Line 160-161 – please add a full text description to “VZV, EBV, CMV, HSV, HHV6, HBV” viruses, as this is the first time, they are referred in the text.

Ethics: the written informed consent for publication obtained from the child parents was validated by any ethical committee (University or hospital)? If so, this should be explained.

Author Response

To reviewer 2:

1) LINE 13: we removed "with" as suggested

2) LINE 160 -161: we added a full text description to " VZV, EBV, CMV, HSV, HHV-6, HBV viruses, as suggested

3) The written informed consent for publication obtained from the child parents was not validated by any ethical committee

Round 2

Reviewer 1 Report

OK.